# Modelling of Urban Air Pollutant Concentrations with Artificial Neural Networks Using Novel Input Variables

**DOI:** 10.3390/ijerph17062025

**Published:** 2020-03-19

**Authors:** Laura Goulier, Bastian Paas, Laura Ehrnsperger, Otto Klemm

**Affiliations:** Climatology Research Group, University of Münster, Heisenbergstraße 2, 48149 Münster, Germany; bastian.paas@uni-muenster.de (B.P.); laura.ehrnsperger@uni-muenster.de (L.E.); otto.klemm@uni-muenster.de (O.K.)

**Keywords:** ANN, prediction, traffic, sound, acoustic, ozone, nitrogen oxides, ammonia, particulate matter, deep learning

## Abstract

Since operating urban air quality stations is not only time consuming but also costly, and because air pollutants can cause serious health problems, this paper presents the hourly prediction of ten air pollutant concentrations (CO_2_, NH_3_, NO, NO_2_, NO_x_, O_3_, PM_1_, PM_2.5_, PM_10_ and PN_10_) in a street canyon in Münster using an artificial neural network (ANN) approach. Special attention was paid to comparing three predictor options representing the traffic volume: we included acoustic sound measurements (sound), the total number of vehicles (traffic), and the hour of the day and the day of the week (time) as input variables and then compared their prediction powers. The models were trained, validated and tested to evaluate their performance. Results showed that the predictions of the gaseous air pollutants NO, NO_2_, NO_x_, and O_3_ reveal very good agreement with observations, whereas predictions for particle concentrations and NH_3_ were less successful, indicating that these models can be improved. All three input variable options (sound, traffic and time) proved to be suitable and showed distinct strengths for modelling various air pollutant concentrations.

## 1. Introduction

The effect of air pollutant exposure on human health has been intensively studied; exposure increases the risk of several diseases for both children and adults [1,2,3]. In particular, nitrogen oxides (NO_x_) and particulate matter (PM) can cause significant long-term health effects such as respiratory diseases, cardiovascular damage, and different types of cancer [3,4], whereas NO_x_ and tropospheric ozone (O_3_) may also have short-term effects on the respiratory system [4]. According to the United States Environmental Protection Agency (USEPA), the pollutants NO_x_, tropospheric O_3_ and PM are the main air components that define the air quality in a certain area [4,5]. Furthermore, NO_x_, tropospheric O_3_, PM, CO_2_ and particle number concentration (PN) are highly important in terms of climate change, leading the EU to commit to complying with the specified limit values for NO_2_, O_3_, PM_2.5_ and PM_10_ [6]. Another pollutant, ammonia, is an atmospheric precursor for fine inorganic aerosol formation, and it is toxic to the environment [7,8]. Recent studies have started investigating whether NH_3_ is emitted from catalytic converters in cars [7,9], as NH_3_ may be emitted from three-way catalytic converters (TWCs) in gasoline-powered cars. In diesel-powered cars, selective catalytic reduction (SCR) uses urea in order to reduce NO emissions, but NH_3_ may still be emitted after thermal decomposition and hydrolysis [9].

Motor traffic is not only the major source for NO_x_ [10] and PM [11] in urban environments, but, due to the noise it creates, traffic also increases people’s stress levels and, consequently, impacts their health [12,13,14]. Especially in street canyons, both the noise levels and the NO_2_ and PM_10_ concentrations are high [15]. As an increased traffic density correlates with a high noise level [16], several studies have shown that urban air pollutant concentrations can be well described by metrics of sound [17,18]. Alternatively, another proxy for the traffic density and, in turn, air pollutant concentrations, can be achieved by combining the time of the day with the day of the week [19].

While sources of air pollutants beyond traffic, e.g., industry, are also important for certain cities, the main drivers of local pollutant concentrations in the urban atmospheric boundary layer are traffic emissions, background concentrations and meteorological conditions that control the transport of pollutants [20]. Thus, these drivers should be considered as predictors in air pollution models. Since environmental relationships are nonlinear and reasonably complex, this context is well suited to artificial neural networks (ANNs) [18,21,22,23]. In particular, the modelling of urban NO_2_/NO_x_ concentrations [24] and PM/PN concentrations [18] using a Multilayer Perceptron (MLP) has shown good prediction results. The basic idea of ANNs is to mimic processes that occur in the human brain. They receive and process information and output results, whereby the network can restructure itself during processing [25]. By recognizing certain patterns within input datasets, they can learn the best way to predict the output [26]. 

This study presents the development and evaluation of ten ANN models for predicting urban air pollutant concentrations (CO_2_, NH_3_, NO, NO_2_, NO_x_, O_3_, PM_1_, PM_2.5_, PM_10_ and PN_10_) using meteorological data, background concentrations and certain predictors of traffic volume, namely sound, traffic and time, as new input variables. We selected these particular novel input variables—acoustic sound measurement data (sound), the total number of vehicles (traffic) and the hour and weekday (time)—in order to identify a simple yet accurate method for prediction. This way, the model approach was simplified by using freely accessible data as much as possible. The ANNs were trained, validated and afterwards tested to evaluate their performance in a comprehensive statistical analysis. Overall, this work represents a feasibility study that evaluates a fast and cost-effective approach for 1 h mean value prediction of pollutant concentrations in order to predict precise urban air pollutant concentrations at moderate costs and rather little computing effort; as such, it potentially offers a toolbox for urban planners.

## 2. Materials and Methods

### 2.1. Site Description

All data were collected in the city of Münster, northwest Germany, from 30 August 2018 through 31 October 2018 (Figure 1). Most of the data were measured in an inner-city road canyon called “Bült” (51°57′49.2″N, 7°37′52.3″E), where the street consists of two traffic lanes, one in each direction, surrounded by buildings of three or four floors and a height of nine to twelve meters (aspect ratio: 0.56–0.75). As the street is used as an important east–west connection right in the city centre, it is a busy street, with a daily traffic volume of over 33,000 vehicles, including coaches of 19 different public bus routes. The bus stop is located 75 m to the NW from the measurement site. The speed limit on Bült is 30 km h^−1^. Traffic jams occur often, especially during the rush hours. Due to a channelling effect in the street canyon, the wind direction is almost exclusively either NW or SE. Data describing the stability of the atmospheric boundary layer were collected at a permanent measurement station of the University of Münster, located at Steinfurter Straße (51°58′48.5″N, 7°35′57.3″E), approximately 2.9 km to the NW of Bült. Background concentrations of the air pollutants were taken from the North Rhine-Westphalian State Office for Nature, Environment, and Consumer Protection (LANUV) at their Münster-Geist Background Station (MSGE) at Oberschlesier Straße 73 (51°56′11.5″N 7°36′41.5″E), approximately 3 km to the south-southwest of the city centre. The measurement campaign was set up on top of an overseas container at a height of 4 m above ground level (AGL).

Figure 2 shows the setup of the measurement station on Bült.

The container was positioned in a parking lot in front of Bült number 16, next to the street (Figure 3). Bült is slightly curved from a WNW–ESE orientation to a NNW–SSE direction. 

### 2.2. Data Measurement and Processing

#### 2.2.1. Meteorological Data

The meteorological data were measured at the Bült station. For wind speed and wind direction, a Gill R3-50 ultrasonic anemometer (4.40 m AGL) with a time resolution of 10 Hz was used. Temperature and relative humidity were measured with a Campbell HC2-S3 air-temperature and humidity sensor (3.54 m AGL) with a time resolution of 1 min. The meteorological data were averaged if at least 75% of the hourly data were valid; this represents the currently requested percentage of the available data for modelling [27]. LANUV also uses this standard as a quality criterion for processing the background concentrations (see Section 2.2.3). At Münster Steinfurter Straße a Gill R3-50 ultrasonic anemometer was operated with 10 Hz time resolution at 10 m AGL over a flat agriculture field. The software package Eddy Pro 6.2.0 [28] was then employed to calculate the friction velocity u* and the stability parameter ζ = z / L (z = height AGL, L = Monin–Obukhov length).

#### 2.2.2. Pollutants

The concentrations of the trace gases CO_2_, NH_3_, NO, NO_2_ and O_3_, as well as four particle metrics, i.e., particle mass concentrations PM_1_, PM_2.5_, PM_10_ and particle number concentration PN_10_, were measured at Bült. To quantify CO_2_ concentrations, a LI-COR LI-7500 A open path CO_2_/H_2_O gas analyser (4.34 m AGL) with a time resolution of 10 Hz was used; for NH_3_ concentrations, a Teledyne M501E NH_3_ analyser (3.78 m AGL) measuring at 2 Hz was employed. The NO, NO_2_, NO_x_ (NO_x_ = NO + NO_2_) and O_3_ concentrations were determined with an EcoPhysics CLD 899 Y for NO, NO_2_, NO_x_ (4.04 m AGL) and an EcoPhysics CLD 88 for O_3_ (4.03 m AGL) at 10 Hz. EcoPhysics as well as Teledyne NH_3_ instruments, which are both based on the principle of chemiluminescence [29], were calibrated in the laboratory before the measurement campaign. Particle mass concentrations as well as particle number concentrations were measured with a Dekati electric low-pressure impactor ELPI+ (3.96 m AGL) and, hence, contain particle information in a range from 6 nm to 10 μm particle diameters. An ELPI+ baseline calibration was done every two to three days after changing the impaction foils and reinstalling the impactor.

#### 2.2.3. Background Data

The LANUV operates background stations in many cities in North Rhine-Westphalia (NRW) to obtain a picture of the urban background concentrations. As Münster is mostly surrounded by agriculture, it is likely that the background reflects an agricultural rural environment. Background concentrations for most of the measured pollutants were accessible from the Münster-Geist Background Station (MSGE). The raw data consisted of 0.2 Hz data for NO, NO_2_, NO_x_, O_3_ and PM_10_ particle mass concentration. For quality control, all data with quality code (FC) > 7 were excluded, and a 1 h average was calculated only if at least 75% of the hourly data were valid. This data quality/quality assurance (QA/QC) procedure (see Section 2.2.1) was determined by the LANUV itself.

#### 2.2.4. Acoustics

Traffic is a major source of noise pollution at Münster Bült [30]. Continuous data for physical sound pressure were recorded with an NTI Sound Level Meter XL2 at 24 Bit [31] and using a Class 1 M2230 microphone with a sensitivity of 43.9 mV Pa^−1^ at 1 kHz. It was calibrated with a Class 1 Precision Calibrator 94/114 at a frequency of 1 kHz in dry conditions, on a daily basis for the first two weeks and then every third day after good calibration values were confirmed. The maximum deviation was at 0.1 dB for the entire period. For the analysis, the equivalent sound pressure level (LZeq) was calculated. By doing so, the integral of unweighted sound levels over the frequencies between 5 Hz and 20 kHz was calculated and temporarily averaged (1 h mean). Paas et al. (2016) found that LZeq was a good acoustic representative for motor vehicle traffic. Thus, LZeq is considered a good predictor for particle emissions within street canyons and, thus, was selected as an input parameter for ANNs. The data processing followed recommendations of Paas et al. [18]. 

#### 2.2.5. Traffic

As traffic is one of the major sources of urban pollutants [4,20], we used the total number of vehicles as an alternative predictor for ANN processing. Two cameras (universe number plate cameras ANPR Long Range 25 mm), one set for each traffic lane, captured information on the traffic volume. These data were anonymously processed by the Federal Motor Transport Authority (KBA) and averaged to 1 h. The required video surveillance in public spaces and the respective data handling had been approved before the experimental period by the responsible data protection officer according to current European Union legislation.

#### 2.2.6. Time

In order to offer very simple input variables to the model, the parameter time was used as a third alternative to train a model for predicting pollutant concentrations. Traffic intensity and more factors that exert an influence on pollutant concentrations typically occur in a recurring temporal pattern [20]. Time of day and the day of the week have previously shown good prediction power [19]. Therefore, information about the hour and whether it is a workday, Saturday or Sunday were included into the input dataset instead of sound and traffic. The time of day and the day of the week can reflect traffic behaviour and rush hours. The time always indicates the beginning of the respective hour, so that “10” stands for the hour between 10 and 11 hrs CET. 

#### 2.2.7. Data Processing

All data that were used for the ANN were measured continuously and averaged to 1 h mean values. In total, the time series included data from 1481 hours. First, a strict QA/QC procedure was applied using R Studio [32]. All data were normalised to the range from 0 to 1 (Equation (1)) [18,20,33].
(1)$$x_{\mathrm{norm}}=\frac{\left(x_{i}-x_{\min}\right)}{\left(x_{\max}-x_{\min}\right)},$$

The normalised hourly average of parameter x (x_norm_) was computed by subtracting the minimum value (x_min_) of the entire time series from the original value (x_i_) and subsequently dividing the results by the difference between the highest (x_max_) and the lowest value (x_min_). This procedure yields datasets that are unbiased by extremes and can therefore be directly processed with an ANN [20,24]. Normalisation was performed individually for each variable in each ANN model. For the statistical performance analyses of the models, all values were transformed back to absolute values.

### 2.3. Multilayer Perceptron

A Multilayer Perceptron (MLP) is a feedforward neural network organised in different layers consisting of different neurons [25,34]. As MLPs showed good performance in predicting air pollutant concentrations [20,21,26], we employed MLPs in this study to model concentrations of ten different air pollutants in a street canyon: CO_2_, NH_3_, NO, NO_2_, NO_x_, O_3_, PM_1_, PM_2.5_, PM_10_ and PN_10_. The information processing flow is not circular but moves in a forward direction (input layer → hidden layer → output layer). The input layer is visible and provides all relevant signals from the external environment. Then, the provided information is processed within one single hidden layer [34], which enables the neural network to model nonlinear relationships between parameters and provides a better generalisation than a single-layer perceptron. The output layer describes the response of the MLP to the questions raised, in this case the prediction of pollutant concentrations [35,36]. All connections between neurons are translated into specific connection weights which quantify the strengths of positive, negative or neutral relationships between information. The greater the weights between the neurons, the stronger their correlations and the stronger their effect on the output signal [25,34,37]. This allows the model to identify and quantify associations between predictors and output variables [34]. 

The design of the model’s architecture is crucial in developing the best possible MLP. The architecture is mostly defined by the number of neurons in the input layer, the number of hidden layers, the number of neurons within the hidden layer, the weights between all neurons, and by the transfer function. The type of this transfer function can widely differ from linear functions to binary normally distributed functions to sigmoid functions. Typically, a sigmoid function is selected. Sigmoid functions reduce the importance of extreme values within a dataset without removing these values. In a logistic sigmoid transfer function, the value range is limited between 0 and 1; for a tangents hyperbolic transfer function, the value range is between +1 and −1 [25,34,35,38]. Regarding the number of hidden neurons, the ideal number is determined during the model structure development process (see Section 2.3.3), whereas the optimum connection weights between neurons are identified during the training process and by using a backpropagation algorithm. The backpropagation algorithm is used for computational error minimisation for feedforward networks [20,25]. At the beginning of the training, all weights are randomly set, and their errors are evaluated. Hereby, all error values are squared and summed up. If the output results are sufficiently precise, the weights are kept as they are. Otherwise, if the errors reach a certain threshold value, they need to be adjusted to reach a better result. In this case, the error values are passed back into the opposite direction towards the input layer. In accordance with the calculated error signal, the weights are gradually modified by the gradient descent, first for the hidden layer and the output layer, and then for the input layer and the hidden layer [20,25]. For neural networks that contain more than one hidden layer, the gradient descent commences for the last hidden layer and the output. This process is repeated for the entire training dataset until the output reaches an error minimum [25]. Further mathematical calculations during the backpropagation process are described in Rojas, 1996 [39]. Figure 4 shows the architecture of the MPL to predict O_3_ concentrations with the time of the day and day of the week (time) as a predictor. The perceptron contains seven input variables in the input layer and two hidden neurones and a hyperbolic tangent transfer function in the hidden layer to model O_3_ concentrations as the output (see Section 2.3.3).

#### 2.3.1. Input Variable Selection

The input variable selection (IVS) is a fundamental step in ANN model development. The aim is to identify the most relevant variables out of the total number of variable candidates available [40,41]. Even though an input variable candidate might have a strong correlation with the output layer, its information can still have little prediction power if another input variable already supplies the same information. Thus, it is important to find the set of input variables with the fewest candidates needed to define the desired output properties. While too little information cannot sufficiently supply all necessary information to predict a good output, redundant and noisy information can lead to a more difficult, slower, and less effective training process with worse model prediction accuracy. Models with redundant variables show less generalisation capacity since the noise covers essential input-output correlations [41,42]. Thus, for adequate IVS, we chose the partial mutual information (PMI) method by Sharma (2000) [43], as this unbiased approach tests the significance of all potential variables and inherently eliminates redundancies [42,44,45]. Further information about the formula and calculation of PMI-based selection (PMIS) can be found in Fernando et al. 2008 [42].

Generally, urban air pollution is driven by the background concentrations, local emissions and their local dispersion [20]. It is important for the ANN that each of these drivers is represented by at least one input variable. The background concentrations are represented by the LANUV suburban background data. To ensure best fitting of background concentration for NO, NO_2_, NO_x_ and O_3_ modelling, another PMIS was separately run to identify the optimal background variables. Thus, no redundant information was included into the PMIS. The emission data are represented by three different variables, which were used alternatively and whose performances were tested and compared to each other; as such, LZeq, n_traffic_, and the hour and day of the week were included into the ANN model development. These parameters were used as direct predictors for traffic emissions without previous PMIS testing. Motor traffic emissions are known as a major source for inner city pollutant concentrations [15], and, even though factors such as industry, domestic fuel, and restaurants also have an influence on the pollutant concentrations, we did not take them into account because measuring these impacts is not considered effective yet is time consuming and inexact [20]. Regarding air pollution dispersion and distribution, these are well accounted for in ANNs by meteorological data [20]. The meteorological variables air temperature (Ta), pressure (p), relative humidity (rH), wind speed (WS), wind direction (WD), u*, and ζ were analysed within a PMIS calculation process. Precipitation is also proven to have a strong impact on pollution dispersion, especially with regard to the wash-out effect for particles [18,20]. However, precipitation was not included in ANN development as these data were not available. All meteorological data as well as background concentrations and time as a direct predictor are available on an area-wide basis and can be used at no additional costs. 

In order to interpret the output of the PMIS analysis correctly, the Akaike information criterion (AIC) was used, as it expresses the best trade-off between accuracy (model error) and the amount of input variables in the input dataset (model size). The AIC automatically punishes overfitting by excluding redundant information. Mathematical information about the AIC can be found in May et al. 2008 [44] and Akaike 1974 [46]. To determine the final set of input variables, all candidates up to a global AIC minimum should be included into the model development [41,44]. The resulting set of input variables for each pollutant is described in Section 2.3.3.

#### 2.3.2. SOM-Based Stratified Data Splitting

Data splitting is a key aspect during ANN development. In order to rule out overfitting, the dataset for each pollutant is divided into three data subsets: a training Ttr, a validation Tv, and a test dataset Tt [40,47,48]. The ANN needs to be robust to provide a good fit to the training set as well as a good accuracy when calculating with unseen data. These two demands lead to a conflict and are referred to as the *Bias and Variance Dilemma* [49]. Overtraining is avoided by the cross-validation technique [35]. This technique takes the majority of the data (70%) for training. Another 20% of the data is used to evaluate the performance during the training. As soon as the performance reaches its best result at a minimum error, the training improvement is stopped [50]. The test data (10%) are used to examine the model performance with previously unseen data [47]. To ensure that the whole variety of data points is represented and the statistical properties are alike in each data subset, all datasets are split by the self-organizing map (SOM) method [51]. This is an unsupervised learning technique with no clearly defined output. Within this method, all data are organised in weighted clusters of the same properties [25]. Similar elements are moved closer together, dissimilar ones further away from each other [34]. Afterwards, the data within each cluster are randomly allocated to the subsets, so that each cluster is represented in each of the subsets. Thus, all samples from each cluster are selected with the same probability. This is a so-called SOM-based stratified random data splitting (SBSS) [40]. Not only the sample allocation but also the size of the SOM is an essential setting to minimise the bias and variance within the datasets. Hereby, the SOM grid size (SOM_gs_) is calculated by the sample size (Sn), where the SOM_gs_ equals Sn^0.54^. The length of the SOM map is 1.6 times the SOM_gs_, and the width is equal to the SOM_gs_. Table 1 shows all SOM parameters set for the SBSS [18,40]. Further mathematical information about the SBSS calculation can be found in May et al. 2008 [40].

All calculated grid sizes for every pollutant are shown in Table 2. 

#### 2.3.3. Model Structure Development

Input variables for each pollutant with the predictors sound, traffic, and time were selected according to a PMIS method as described in 2.3.1. In order to detect the optimum number of hidden neurons, a trial and error principle was applied, beginning with one single neuron within the hidden layer. Adding one neuron at a time, up to 20 hidden neurons per pollutant were tested to find the optimum number. Meanwhile, the correct transfer function had to be identified. Two transfer functions (hyperbolic tangent and logistic sigmoid) were used for each of the 20 numbers of hidden neurons. Parameters set for calculating the MLPs are listed in Table 3. In this study, the learning rate was set to 0.1 and multiplied by the error signal in order to adjust the connection weights during the backpropagation process. In general, high learning rates enlarge the steps of the weight correction, while small values reduce it. A larger step saves time but carries the risk that the minimum is missed. In this case, the backpropagation algorithm jumps back and forth around the minimum without ever reaching it [34]. The momentum parameter prevents the learning algorithm from getting stuck in certain search spaces; therefore, it helps to find the minimum and supports better model quality [34]. Within this study, a momentum value of 0.9 was used.

For each combination of hidden neurons and transfer function, the root mean squared error (RMSE) of the validation dataset was calculated, and the combination with the lowest error was selected as the optimum architecture [18,23]. If two RMSE values were identical or significantly close, the standard deviation (SD’) was additionally calculated and chosen to be the deciding parameter. All results for model architectures for each pollutant with the predictors sound, traffic and time are presented in Table 4. 

### 2.4. Analysis of Model Performance

For analysing the model performances, several statistics were used, since all of them have their own advantages and drawbacks [27]. For the analysis of model fitness, we included the root mean squared error (RMSE), the correlation coefficient (R), the normalised mean standard deviation (NMSD), the normalised mean bias (NMB), the rank correlation by Spearman (r_s_), and the standard deviation of the observed values (SD) as well as the modelled values (SD’). The Forum for Air Quality Modelling in Europe (FAIRMODE) has proposed the RMSE, R, NMB and NMSD as a core set of statistical indicators for assessing the performance of air quality models [27]. The RMSE (Equation (2)) was used during the model structure development as well as for final assessment of the model fitness, as it provides general information on the deviation between a modelled value and an observed value. Hereby, M_i_ are the modelled values, and O_i_ the observed ones. Their differences are squared and summed up, then square-rooted and divided by the total number of values.
(2)$$\mathrm{RMSE}=\frac{1}{N}\sqrt{\sum_{i=1}^{N}{\left(M_{i}-O_{i}\right)}^{2}},$$

In the context of the FAIRMODE’s Guidance Document on Modelling Quality Objectives and Benchmarking, R has been calculated with Equation (3), NMB with Equation (4) and NMSD with Equation (5) [27]. NMB indicates how much a model over- or underestimates the observed values, where positive values describe an overestimation, negative values an underestimation. The closer the NMB is to zero, the smaller the over- or underestimation. Additionally, the r_s_ was calculated, as it uses the rank of data and arranges the values in an ascending order before calculating the relationship between the observed and the predicted values [52]. The r_s_ value can range from 0 to 1. The closer the value to 0, the weaker the correlation between the ranks.
(3)$$R=\frac{\sum_{i=1}^{N}\left(M_{i}-\bar{M}\right)\left(O_{i}-\bar{O}\right)}{\sqrt{\sum_{i=1}^{N}{\left(M_{i}-\bar{M}\right)}^{2}}\sqrt{\sum_{i=1}^{N}{\left(O_{i}-\bar{O}\right)}^{2}}},$$
(4)$$\mathrm{NMB}=\frac{\mathrm{BIAS}}{\bar{O}},$$
(5)$$\mathrm{NMSD}=\frac{\sigma_{M}-\sigma_{O}}{\sigma_{O}},$$

To assess whether the model performs sufficiently well, the FAIRMODE’s Guidance Document provides the Modelling Quality Indicator (MQI), a single value that either considers the results acceptable or not. Therefore, it divides the RMSE by the measurement uncertainty (RMS_U_) and a fixed factor (β) as shown in Equation (6).
(6)$$\mathrm{MQI}=\frac{\mathrm{RMSE}}{\beta {\mathrm{RMS}}_{U}},$$

The fixed factor β is set to 2 by the FAIRMODE, allowing the difference between the modelled and the observed values to be twice the measurement uncertainty. The RMS_U_ calculation is shown in Equation (7).
(7)$${\mathrm{RMS}}_{U}=\sqrt{\frac{\sum_{i=1}^{N}{\left(U\left(O_{i}\right)\right)}^{2}}{N}},$$

For calculating the measurement uncertainty of the observed values (U(O_i_)), we used Equation (8). Within this equation, α is a fraction between 0 and 1, set by the FAIRMODE itself, as well as values for the relative standard measurement uncertainty around the reference value (U_r_(RV)) and the reference value (RV). The actual values for α, U_r_(RV) and RV for calculating U(O_i_) for NO_2_, O_3_, PM_10_ and PM_2.5_ can be found in Janssen et al. (2019) [27].
(8)$$U(O_{i})=U_{r}\left(\mathrm{RV}\right)\sqrt{\left(1-\alpha^{2}\right){O_{i}}^{2}+\alpha^{2}{\mathrm{RV}}^{2}},$$

## 3. Results

Concentrations of ten air pollutants (CO_2_, NH_3_, NO, NO_2_, NO_x_, O_3_, PM_1_, PM_2.5_, PM_10_ and PN_10_) have been modelled using an MLP. For modelling, the equivalent sound pressure level (LZeq), the total number of vehicles and the hour and day of the week were compared as three different input variable options representing the motor traffic intensity. Meteorological data and background concentrations were also included in the IVS. For analysing the ANN performance, observed and modelled values were compared, with each data point representing a 1 h average. Figure 5 shows the results for the trace gases CO_2_, NH_3_, NO, NO_2_, NO_x_ and O_3_ comparing the three input variable options. All models, except NH_3_, show a positive correlation between the predictions and the observations for all three input variable options. Using all three input variable options, NH_3_ concentrations were predicted around the mean of observations without much variation. The model is completely insensitive to actual changes in the input layer. Both CO_2_ and NO show fairly good prediction results. Results show a clear positive correlation between modelled and observed values. Values in the high-end and the low-end concentration range scatter most. Intermediate values show good agreement. NO_2_ and O_3_ models perform best, having a very small variance and a good fit to the 1:1 line (see Figure 5). NO_x_ results are fairly good but cannot supply the same precision as predictions of NO_2_. The NO_x_ model using sound data instead of traffic counts or time data as input variables shows the best results. 

Figure 6 shows the results for aerosol particle predictions. Throughout, the modelled particle concentrations show worse agreement with the observations than the models for trace gases (see Figure 5). Models for PM_1_ using sound, traffic and time as input variable options do not show any reaction to changing inputs. Only an average value was calculated without much variation, especially using traffic input data. Predictions of PM_2.5_ show moderate accuracy for modelling with sound input data, whereas the model using traffic input data is completely insensitive to changing inputs. For the model using time input data, no positive correlation between the modelled and observed values could be identified. The model results scatter widely around the mean value of observations. Modelling PM_10_ with input variable options of both traffic and time shows a positive correlation, although the variance is very large. The model using sound input data is insensitive to changes in the input layer. The models predicting PN_10_ concentrations using both sound and time input variable options show a slightly positive correlation with the observed values with a large variance. Overall, the results for all three input variable options outline poor performance for PN_10_ modelling. 

To determine which of the input variable options (LZeq, n_traffic_, hour and weekday) performs best in predicting pollutant concentrations, the RMSE was calculated for comparison. In Figure 7, which shows normalised RMSE values for all pollutants, one can see that the best RMSE values emerge for three models when using sound input data (NO_x_, PM_2.5_, PN_10_) for four models when using traffic data inputs (CO_2_, NO, PM_1_, PM_10_) and for three models when using time input data (NH_3_, NO_2_, O_3_). 

In addition to the RMSE, Table 5 shows other statistical parameters for evaluating model performance (see Section 2.4). Through the evaluation of R, good prediction accuracy can be confirmed for models of NO using traffic inputs (R of 0.739), NO_2_ across all input data (R of 0.878, 0.812 and 0.915), NO_x_ with sound input data (R of 0.751), and O_3_ across all input data (R of 0.750, 0.821 and 0.871). Values of R point out moderate prediction accuracy for CO_2_ across all input data (R of 0.569, 0.685 and 0.658), NH_3_ using sound input data (R of 0.648), NO_x_ using both traffic and time inputs (R of 0.646 and 0.698), PM_2.5_ with sound (R of 0.587) and PM_10_ with traffic input data (R of 0.536). While the evaluation of R shows poor model performance for both NH_3_ and PM_2.5_ using traffic (R of 0.233 and 0.353) and time input data (R of 0.329 and 0.163), the models of PM_1_ and PN_10_ show poor performance using all three input variable options (R of 0.315, 0.240 and 0.384 for PM_1_ and R of 0.449, 0.253 and 0.397 for PN_10_). For the models of both NO and PM_10_ using sound (R of 0.478 and 0.400) and time inputs (R of 0.346 and 0.425), poor performance was discovered. The rs values support the above findings for CO_2_ (r_s_ of 0.616, 0.695 and 0.665), NH_3_ (r_s_ of 0.632, 0.239, 0.403), NO (r_s_ of 0.531, 0.751 and 0.390), NO_2_ (r_s_ of 0.816, 0.830 and 0.881), NO_x_ (r_s_ of 0.761, 0.703 and 0.685), O_3_ (r_s_ of 0.737, 0.811 and 0.870), PM_1_ (r_s_ of 0.188, 0.289 and 0.416), PM_2.5_ traffic and time (r_s_ of 0.192 and 0.149), PM_10_ (r_s_ of 0.148, 0.658 and 0.519) and PN_10_ (r_s_ of 0.524, 0.249 and 0.445). In contrast to the R values above, the rs value for PM_2.5_ sound (r_s_ of 0.384) shows poor model results. The NMB points out that 19 out of 30 models underestimate the observed values, while 10 overestimate them. For CO_2_ using traffic input data, the NMB equals 0.000. Generally, none of the three predictors consistently show the best result for all gases or particle measures.

While all statistical parameters have advantages and disadvantages with respect to the evaluation of models, the MQI was developed by the FAIRMODE to find the limit of acceptable performance with a single number. Table 6 shows the MQI for the models of NO_2_, O_3_, PM_2.5_ and PM_10_. For all other models, MQI calculations could not be performed due to the lack of available reference values from the FAIRMODE. All models reaching MQI values lower than 1 can be considered sufficiently good. The models of both NO_2_ and O_3_ show very good results for all three input variable options. Concentrations of NO_2_ and O_3_ are best predicted with time input data (MQI of 0.281 and 0.255). Models using traffic input data show the second best results (MQI of 0.342 and 0.327). Although models using sound input data show the worst MQI for the prediction of NO_2_ and O_3_ concentrations, the performance is still very good (MQI of 0.357 and 0.414). The MQI values for PM_2.5_ models present moderate but still acceptable performance with all three input variable options (MQI of 0.566, 0.778 and 0.828) even though the models using traffic and time input data do not reveal sensitivity to changing inputs (see Figure 6). The PM_10_ models using sound and time input data have MQI values higher than 1 (MQI of 1.004 and 1.071), which is considered unsatisfactory. The PM_10_ model using traffic input data performs adequately, with an MQI below yet close to 1 (MQI of 0.882).

## 4. Discussion

ANN modelling for the ten air pollutant concentrations (CO_2_, NH_3_, NO, NO_2_, NO_x_, O_3_, PM_1_, PM_2.5_, PM_10_ and PN_10_) using sound, traffic and time data as additional input variable options to represent the traffic volume shows very disparate results for an urban street canyon in Münster, Germany.

Concerning MQI, the performance of all models for which an MQI could be calculated was rated as sufficiently good, except for the PM_10_ model using sound and time input data. The MQI values also reflect the precision of the model, which could also be highlighted with other statistics, showing that NO_2_ and O_3_ give the best results expressed by the lowest MQIs of 0.281 and 0.255, respectively. However, PM_2.5_ models using traffic and time inputs show no response to changes in the input layer even though their MQIs are within the acceptable range (see Figure 6). This leads to the presumption that, at least in PM_2.5_ modelling, the MQI alone cannot adequately assess the model’s performance in this study. Hence, we propose an MQI limit value of 0.7 instead of 1.0 for the analysis of the data used in this study so that models that can only reproduce mean values of observations can be identified more easily through the MQI. 

The modelling of trace gases with ANNs has yielded very precise predictions. A direct link between concentrations of NO_2_, O_3_, CO_2_ and the traffic volume was shown. As the (photo-)chemical reactions between NO, NO_2_ and O_3_ in the atmosphere cause a strong relation between these gases [53], it is expected that modelling NO, NO_2_ and O_3_ will lead to a similar quality of the results. Other studies using ANN models to predict concentrations of nitrogen oxides and O_3_ also show good results [19,20,54]. Cai et al. achieved an RMSE value of 21.1 µg m^−3^ for an ANN with NO_2_ background concentrations as input variables, which is slightly worse than the best RMSE obtained in this study (7.288 µg m^−3^ with time input data). For the prediction of O_3_, they arrived at an RMSE of 9.5 µg m^−3^, whereas this study revealed a best RMSE value of 9.031 µg m^−3^ using time input data [20]. Kamali et al. reached model performances with consistently higher RMSE values for predictions of NO, NO_2_, NO_x_ and O_3_ concentrations [19]. 

Particle modelling, on the other hand, shows results that are still in need of improvement considering the RMSE, r_s_, R and MQI. Modelling of PM_2.5_ with sound input data and PM_10_ with traffic input data show acceptable results concerning the statistical performance measures. However, modelling PM_1_ and PM_2.5_ using traffic and time inputs and modelling PM_10_ using sound and time inputs as well as PN_10_ cannot adequately reproduce the emission and transport processes taking place in the atmosphere. Figure 6 reveals that either these models scatter around a mean concentration of observations without reacting to actual changes in the input layer (PM_1_, PM_2.5_ using traffic and time inputs, PM_10_ using sound inputs and PN_10_ using traffic inputs) or the models show a clear positive reaction to changed concentrations but are associated with high uncertainty. Their prediction accuracy was assessed to be poor. For the PM_10_ prediction, the MQI clarifies that sound and time input variables do not produce satisfactory results. Models for the prediction of PM_10_ using traffic inputs and PM_2.5_ using traffic and time input data also reach relatively high MQI values close to 1.

In general, there is considerable room for improvement in modelling particle concentrations using ANNs. Not only are the sources of particles complex, but they also have large within-atmosphere dynamics. For example, impaction and coagulation modify their concentrations and sizes [53]. Car traffic is a source of both primary and secondary aerosol particles [55,56] and the limited dispersion within a street canyon leads to relative accumulation of particles. During low-traffic times, e.g., at night, the diameters increase while the number concentration (PN) decreases [57,58]. Further, particles are scavenged from the atmosphere during rain events [59]; in this case, the number of particles is expected to decrease while the sound volume increases. For very small particles (PM_1_ and PN_10_, which are both dominated by small particles), processes other than traffic presumably drive the particle concentration to a larger extent, so that neither LZ_eq_, n_traffic_ nor time reflects local processes well enough. Those processes may be the transport of secondary particles after nitrate or ammonia formation from the agricultural precursors as well as particle emissions from households [60]. In addition, background concentrations were only available for PM_10_ but not for PM_1_, PM_2.5_, or PN_10_. The PMIS (see Section 2.3.1) has shown for all models that the use of background concentrations as input variables are of very high importance for precise modelling. Results of Cai et al. 2009 [20] also stress the relevance of background concentration data. 

In general, the predictive capacity of our particle concentration models differs from that found in other studies. For example, Bai et al. 2016 show very good results using a backpropagation neural network for modelling PM_10_ concentrations (RMSE of 23.6 µg m^−3^), but we could not achieve those results here (RMSE of 50, 57 and 78 µg m^−3^) [54]. The model accuracy of Paas et al. 2017 to predict aerosol particle concentrations also could not be reproduced here [18]. This could have arisen for reasons. Paas et al. 2017 simplified the dataset for model development in various ways. They avoided rainy conditions during data collection, while in this study data from all weather conditions were used over a two-month period. Thus, washout effects occurred during this work that likely led to spurious effects in the input layer of the ANN and thus to less precise predictions. Whereas data from Paas et al. were averaged over 10 minutes, all data in this study were averaged to 1 h values. This may have led to a loss of information about processes that typically occur on smaller temporal scales and thus to lower correlations between the traffic volume and particle concentrations. Further, the good model performance in Paas et al. 2017 was reached with data taken at different locations with varying distances to vehicle traffic sources, so that a strong spatial gradient was established there; by contrast, this study used data from a fixed site. 

Regarding NH_3_ emissions, our approach could not confirm the hypothesis that NH_3_ is emitted by car catalytic converters; the models show no reaction to changes in the input layer. Values of R between 0.233 and 0.329 and r_s_ values of 0.239 and 0.403 for modelling with traffic and time input data, respectively, reveal very poor correlation between modelled and observed values. Statistical parameters for the model using sound input data indicate a slightly better performance, but they do not support satisfactory model accuracy. The model using traffic input data has the lowest prediction power for NH_3_ when considering r_s_ and R, which indicates that the observed NH_3_ concentrations are mainly influenced by sources other than car traffic. Since the data were averaged over a period of one hour, the car catalytic emission of NH_3_ cannot be ruled out completely as an emission source within our dataset. However, it is more likely that a larger fraction of NH_3_ originates from background transport due to intensive agricultural economy in the surrounding area, at least at the research site and time period of this study.

Ozone is formed in the atmosphere through photochemical reactions. Further, the O_3_ concentrations are strongly driven by the boundary layer stability during the night and mixing during the day [61]. All these processes are of diurnal nature, which is reflected by the good capability of the parameter time to predict O_3_. The nitrogen oxides NO, NO_2_, and NO_x_ (which is the sum of NO and NO_2_) exhibit a complex pattern, which is generally difficult to model. The traffic emissions of NO are well reflected by the high capability of the parameter traffic to predict them. Nitrogen dioxide is formed through a reaction of NO with O_3_, which is reflected by its good predictability by the parameter time. In that sense, their sum (NO_x_) should be well predicted by traffic. Surprisingly, the ANN does not support that hypothesis. It remains unclear why sound is the better predictor as compared to traffic. Likely, the boundary layer dynamics plays an important yet not recognized role.

Limited model performance may also result from methodological limitations or simplifications. The model was kept as simple as possible and aimed to use freely accessible data as much as possible. Furthermore, the campaign to measure the concentrations of pollutants was limited to only two months. Note that the larger the size of the training dataset, the more precise an ANN model will predict previously unseen data. 

Generally, ANN models can often represent environmental relationships with surprising accuracy, although they are not fully understood by traditional theory. Ultimately, all ANNs are based on the intrinsic principle of the “black-box” [24,33]. Hence, it is not possible to trace and adjust individual steps within the model calculation. Connections and weighting adjustments are not visible for the modeler [33].

Future work on the improvement of the ANN performance will have to integrate rain into the IVS as an input variable. Rain is important in two aspects. First, it leads to scavenging of particles and water-soluble gases and thus to smaller concentrations of these. Secondly, it produces sound, an effect that has to be taken into account when working with LZeq data. 

The inner-city street canyon Bült represents a very specific, rather isolated measurement site, where any influences from industry and agriculture was at a minimum. It was thus reasonable to consider traffic as the main source for the variability of air pollutant concentrations. Further, the measurement was carried out in summer, so that domestic fuel is of little importance. For generally valid statements and to cover seasonal variations, long-term measurements throughout the year should be made [19]. 

The three input variable options examined do not show a clear trend of which has the highest predictive power for modelling air pollutants. All three options calculate similar results and have different strengths for different pollutants. They all show good potential as input variables. For future applications, it will be reasonable to look at what resources are available as input variables. For example, data on traffic counts is not available in many places, whereas sound measurement devices are affordable and easy to operate. Further, the time and day of the week are permanently available information that have no cost. However, while sound and traffic data are directly related to absolute traffic volumes, the time data depend on the location’s specific traffic pattern. Therefore, models using time data as a predictor must be re-trained for each location, while models using sound or traffic data can be used without re-training if the values are within the same range as the training dataset.

## 5. Conclusions

Air pollutant concentrations in an urban street canyon were modelled using an ANN with the input of meteorological data, background concentrations and three different traffic volume predictors. The models were developed, tested and evaluated with a dataset from a street canyon in Münster containing the trace gases CO_2_, NH_3_, NO, NO_2_, NO_x_, O_3_, and the particle concentrations PM_1_, PM_2.5_, PM_10_ and PN_10_. The equivalent sound pressure level (LZeq), the total amount of vehicles (n_traffic_), and the hour of the day and day of the week (time) were compared to determine the best traffic volume predictor for modelling each pollutant. Results for NO_2_ and O_3_ conclude a very accurate prediction power of the models with all three input variables, showing nearly perfect MQI, R and r_s_ values when compared to observations. The NO_x_ model using sound input data also shows good model results. Moderate results could be achieved for all CO_2_ and NO models, the NO_x_ model using traffic and time input data, the PM_2.5_ model with sound and the PM_10_ model with traffic input data. They all present a clear positive correlation between modelled and observed values but a lower variation than the observed values. We conclude that for the above-mentioned pollutants, the ANN approach applied within this study is a powerful tool for air pollution modelling. Nonetheless, models predicting the concentrations of NH_3_, PM_1_, PM_2.5_ with traffic and time input data, PM_10_ models using sound and time inputs and all PN_10_ models do not show satisfactory model results, leaving room for model improvement.

For assessing the ANN performance, the MQI represents a suitable parameter for the statistical analysis. Nevertheless, we would suggest a limit value of 0.7 instead of 1.0 for this study to more easily detect models that only reproduce mean values of observations. Since precipitation data were not available as a meteorological input parameter, this data could be considered to possibly improve particle concentration modelling in the future. This may be realised by excluding rain events or by filtering out frequencies within LZeq attributed to rain. Alternatively, rain may be measured in future studies and integrated into the IVS as an input variable.

This study presents modelling results limited to nowcasting. As a next step, one could investigate how capable the model architecture developed with this work is with forecasting. Input variables from weather forecast models could be used as input variables that represent the transport of pollutants. Further, although background concentrations have a large impact within IVS, such data were unfortunately not available for CO_2_, NH_3_, PM_1_, PM_2.5_, and PN_10_. Overall, we can conclude that each input variable option has good potential as a traffic volume predictor within pollutant concentration modelling. Three models showed the best results when using acoustic input data, four when using traffic input data and three when using time input data. Consequently, neither sound, traffic, nor time can be recommended as the best option for predicting all air pollutant concentrations. Future applications of the ANN approach presented in this study can thus use information that is accessible as input variables representing the traffic volume. 

## Figures and Tables

**Figure 1 ijerph-17-02025-f001:**
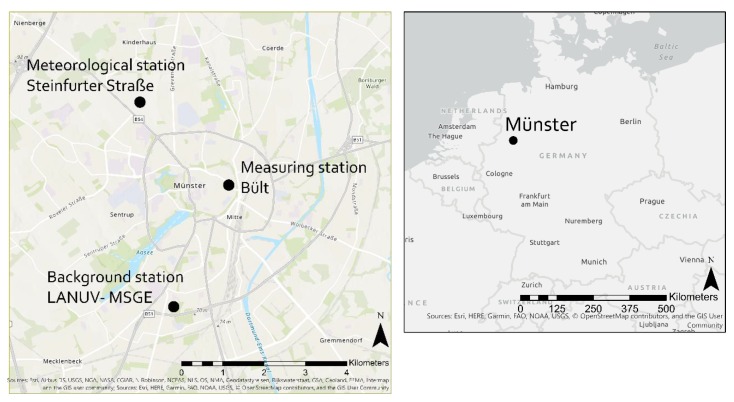
The location of Münster in Germany (**left**) and the three measurement stations within Münster (**right**).

**Figure 2 ijerph-17-02025-f002:**
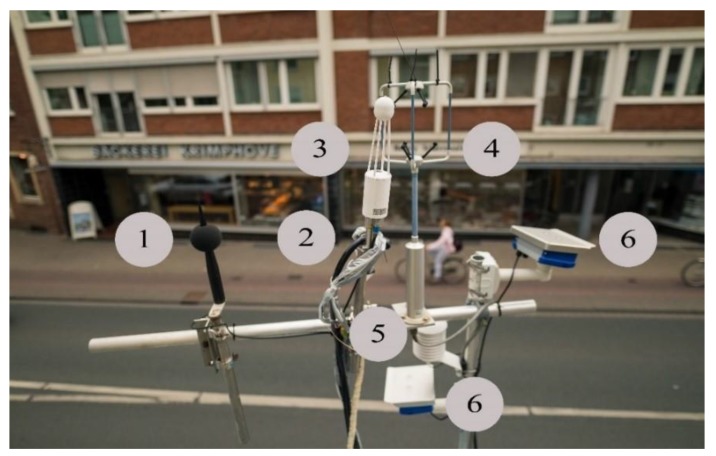
Setup on top of the container. A Class 1 microphone (1), air sample inlets for NH_3_, NO_x_, O_3_, and particles (2), a CO_2_-/H_2_O gas analyser (3), a sonic anemometer (4), a temperature and humidity sensor (5) and traffic cameras one in each direction (6).

**Figure 3 ijerph-17-02025-f003:**
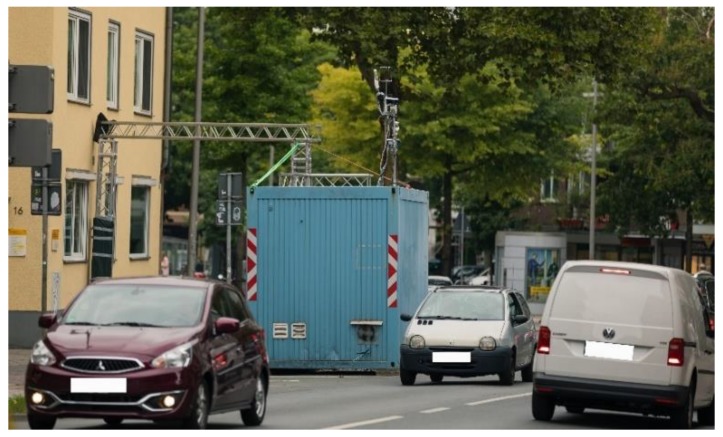
The container hosting the measurement station on Bült in a parking space in front of number 16. The container is equipped with measuring instruments on top.

**Figure 4 ijerph-17-02025-f004:**
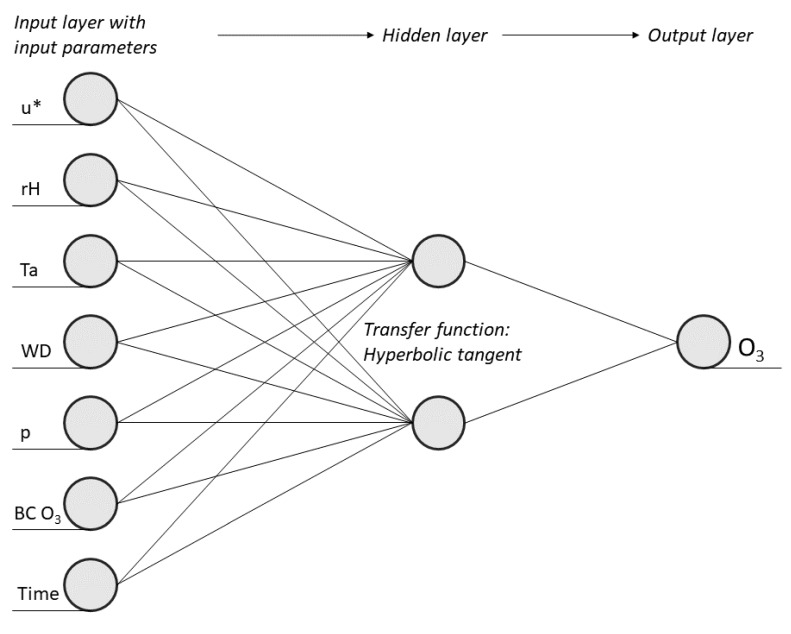
Architecture of the Multilayer Perceptron (MLP) to predict O_3_ concentrations. BC = background concentration.

**Figure 5 ijerph-17-02025-f005:**
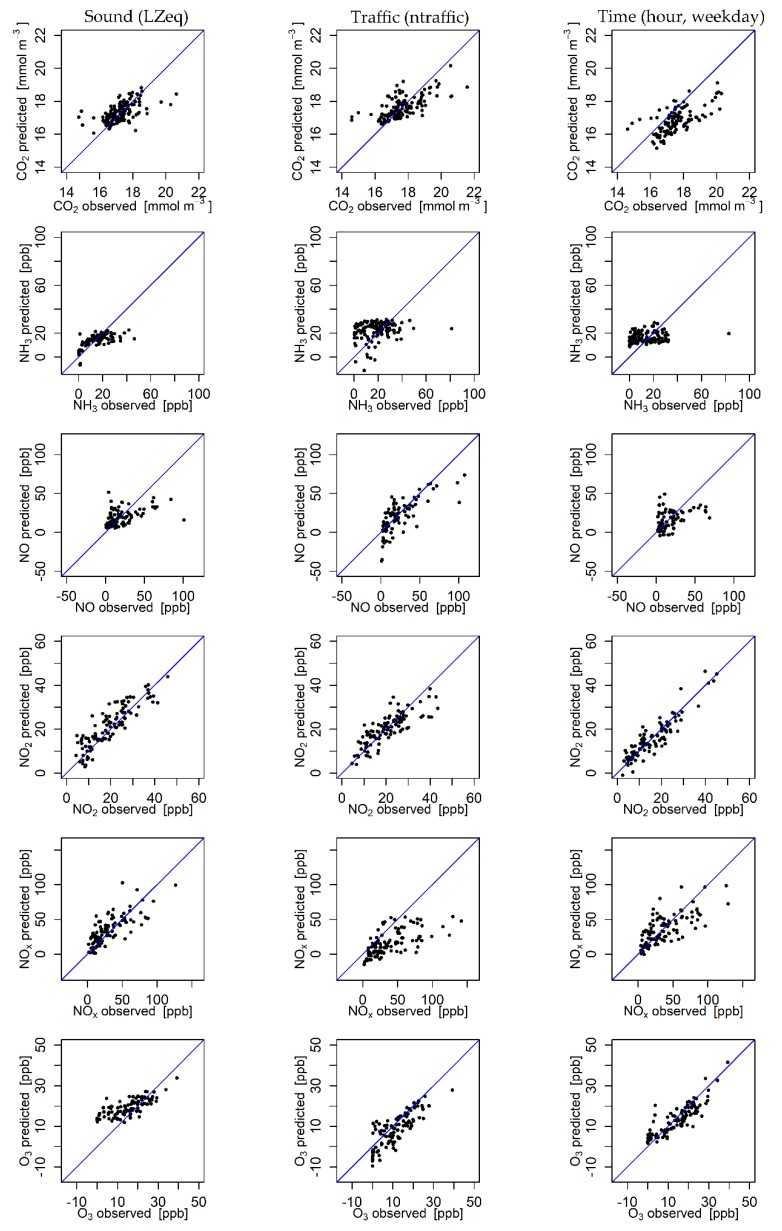
Scatter plot diagrams of trace gas predictions and observations for three input variable options (sound, traffic and time). The blue line illustrates a 1:1 match between the modelled and observed values.

**Figure 6 ijerph-17-02025-f006:**
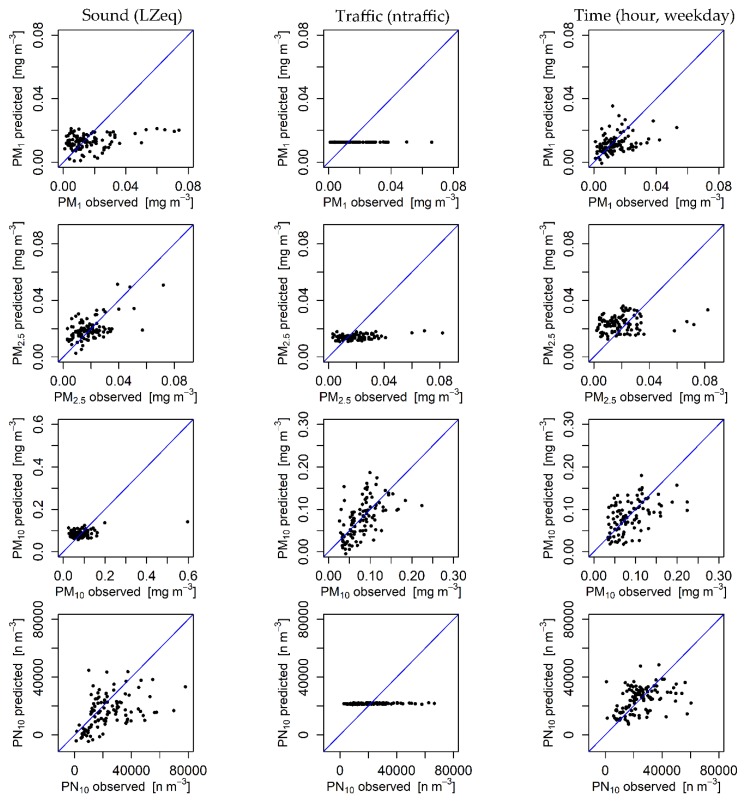
Scatter plot diagrams of aerosol particle predictions and observations for three input variable options (sound, traffic and time). The blue line illustrates a 1:1 match between the modelled and observed values.

**Figure 7 ijerph-17-02025-f007:**
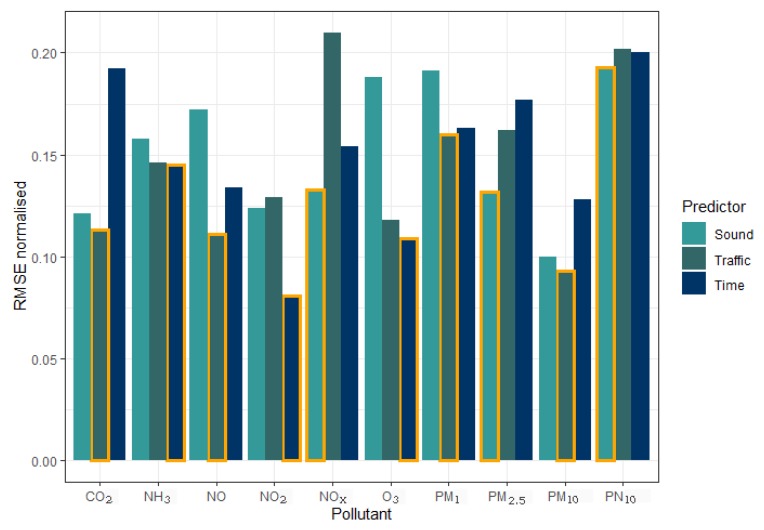
Normalised RMSE values for the prediction of pollutant concentrations (CO_2_, NH_3_, NO, NO_2_, NO_x_, O_3_, PM_1_, PM_2.5_, PM_10_, and PN_10_) with sound, traffic, and time data. Orange framed bars indicate the smallest error for each pollutant.

**Table 1 ijerph-17-02025-t001:** Parameters for the self-organizing map (SOM)-based stratified random data splitting (SBSS) settings.

Parameter	Ordering	Tuning
Initial Learning Rate	0.9	0.1
Initial Neighbourhood Size	See Table 2	1
Epochs	2	20

**Table 2 ijerph-17-02025-t002:** Calculated SOM grid sizes (SOM_gs_) and sample sizes (Sn) for each pollutant.

Pollutant	Sound		Traffic		Time	
	Sn	Width	Sn	Width	Sn	Width
CO_2_	717	10.046	783	10.535	824	10.829
NH_3_	558	8.774	828	10.858	847	10.992
NO	566	8.841	574	8.909	593	9.067
NO_2_	643	9.742	650	9.527	669	9.677
NO_x_	644	9.48	652	9.543	671	9.692
O_3_	644	9.48	652	9.543	671	9.692
PM_1_	658	9.591	651	9.535	687	9.816
PM_2.5_	658	9.591	651	9.535	687	9.816
PM_10_	646	9.496	634	9.400	670	9.685
PN_10_	658	9.591	651	9.535	687	9.816

**Table 3 ijerph-17-02025-t003:** Parameters for Multilayer Perceptron (MLP) calculation for all pollutants.

Parameter	Value
Learning rate	0.1
Momentum	0.9
Max. iterations	100,000
Tolerance	0.0001

**Table 4 ijerph-17-02025-t004:** Model architecture for each pollutant.

Pollutant	Input Variables+LZeq/n_traffic_/time	Transfer Function, Hidden Neurons (Sound)	Transfer Function, Hidden Neurons (Traffic)	Transfer Function, Hidden Neurons (Time)
CO_2_	Ta, p, rH, u*	H.t. 6	L.s. 2	L.s. 9
NH_3_	Ta, p, rH	H.t.15	H.t. 1	L.s.1
NO	Ta, WD, BC NO_x_	H.t.6	H.t. 10	H.t. 10
NO_2_	Ta, WS, WD, rH, BC NO_2_	L.s. 3	H.t. 11	H.t. 17
NO_x_	Ta, WD, BC NO_x_	L.s. 12	L.s. 5	H.t. 19
O_3_	u*, rH, Ta, WD, p, BC O_3_	H.t. 1	H.t. 9	H.t. 2
PM_1_	u*, p, Ta, rH, WD	H.t. 1	L.s. 1	L.s. 6
PM_2.5_	U*, Ta, rH, p, WD	H.t. 8	L.s. 3	H.t. 1
PM_10_	Ta, rH, WD, BC PM_10_	H.t. 8	H.t. 5	H.t. 10
PN_10_	Ta, WS, WD	L.s. 6	L.s. 4	H.t. 7

BC = background concentration (taken from LANUV North Rhine-Westphalia; H.t. = hyperbolic tangent; L.s. = logistic sigmoid.

**Table 5 ijerph-17-02025-t005:** Statistical parameters for evaluating the ANN performance.

Pollutant		RMSE	rs	SD’	SD	NMB	R	NMSD
**CO_2_** [mmol m^−3^]	**Sound**	0.716	0.616	0.558	0.868	0.003	0.569	−0.357
	**Traffic**	0.790	0.695	0.656	1.082	0.000	0.685	−0.394
	**Time**	1.090	0.665	− **	1.066	−0.042	0.658	−
**NH_3_** [ppb]	**Sound**	8.314	0.632	5.540	10.559	−0.118	0.648	−0.475
	**Traffic**	13.474	0.239	7.667	12.427	0.205	0.233	−0.383
	**Time**	11.968	0.403	4.307	12.073	0.312	0.329	−0.643
**NO** [ppb]	**Sound**	17.207	0.531	10.395	19.661	−0.014	0.478	−0.471
	**Traffic**	16.017	0.751	21.107	22.433	−0.133	0.739	−0.059
	**Time**	21.893	0.390	12.078	22.799	−0.161	0.346	−0.470
**NO_2_** [ppb]	**Sound**	5.240	0.816	0.548 *	9.496	0.092	0.878	−
	**Traffic**	5.092	0.830	0.422 *	8.732	−0.017	0.812	−
	**Time**	3.811	0.881	9.115	9.416	−0.002	0.915	−0.032
**NO_x_** [ppb]	**Sound**	16.683	0.761	20.706	24.359	0.133	0.751	−0.150
	**Traffic**	32.820	0.703	17.756	29.728	−0.594	0.646	−0.403
	**Time**	19.983	0.685	21.021	27.848	−0.045	0.698	−0.245
**O_3_** [ppb]	**Sound**	7.372	0.737	4.168	1.828 *	0.294	0.750	1.280
	**Traffic**	5.774	0.811	0.722 *	8.211	−0.271	0.821	−
	**Time**	4.526	0.870	7.506	9.081	−0.057	0.871	−0.173
**PM_1_** [mg m^−3^]	**Sound**	0.014	0.188	0.551 *	0.014	−0.204	0.315	−
	**Traffic**	0.010	0.289	− **	0.010	−0.048	0.240	−
	**Time**	0.009	0.416	0.006	0.009	−0.141	0.384	−0.333
**PM_2.5_** [mg m^−3^]	**Sound**	0.009	0.384	0.008	0.011	0.049	0.587	−0.273
	**Traffic**	0.013	0.192	0.002	0.012	−0.283	0.353	−0.833
	**Time**	0.014	0.149	− **	0.013	0.227	0.163	−
**PM_10_** [mg m^−3^]	**Sound**	0.057	0.148	0.016	0.062	0.066	0.400	−0.742
	**Traffic**	0.050	0.658	0.740 *	0.057	−0.067	0.536	−
	**Time**	0.078	0.519	0.538 *	0.086	−0.131	0.425	−
**PN_10_** [n m^−3^]	**Sound**	15,915.23	0.524	0.922 *	14,946	−0.318	0.449	−
	**Traffic**	12,872.74	0.249	375.384	13,022.7	0.002	0.253	−0.971
	**Time**	11,934.47	0.445	8858.02	12,321.3	−0.002	0.397	−0.281

Values marked with * are logarithmised to guarantee normal distribution. Values marked with ** could not achieve normal distribution by logarithmising.

**Table 6 ijerph-17-02025-t006:** Model Quality Index (MQI) of NO_2_, O_3_, PM_2.5_ and PM_10_ developed by Forum for Air Quality Modelling in Europe (FAIRMODE). Values greater than 1 represent a model performance that is not sufficiently good.

NO_2_			O_3_			PM_2.5_			PM_10_		
Sound	Traffic	Time	Sound	Traffic	Time	Sound	Traffic	Time	Sound	Traffic	Time
0.357	0.342	0.281	0.414	0.327	0.255	0.566	0.778	0.828	1.004	0.882	1.071

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
