# Peer review of "Modelling of Urban Air Pollutant Concentrations with Artificial Neural Networks Using Novel Input Variables"

_ijerph, 2020, doi:10.3390/ijerph17062025_

Round 1
Reviewer 1 Report
I commend the authors on a tremendous paper. It reads like a dissertation with the amount of work and detail. There are only minor things that I would suggest to make it really stand out. I suggest that you have a supplemental section to place the sections where you go into large technical detail about the model and the calculations and how they are done and analyzed. This will allow you to really drive the read through the story without them being too lost in the detail. If they want to know more then they can go to the supplemental section to get that. I think that what you have done is so great that you want the reader to get through the science quicker because that will have a more lasting impact. The only grammar thing that I found was on line 182 where you have hours' you do not need that ' after. Overall I think you guys did a great job trying to simply modeling so that non regulatory people can make decisions easier and probably quicker. Keep at the other two parameters to make them better because I think you have something great!Author Response
Thank you very much for your evaluation. Please see the attachment.

Reviewer 2 Report
In this work, the authors use a suite of instrumentation to measure concentrations of NO, NO2, NOx, O3, NH3, CO2, and PM1, PM2.5, PM10, and PN10 as well as meteorological data (temperature, wind speed, wind direction relative humidity, etc.). This is measured at two sites – in downtown Muenster in a street canyon, and at the LANUV background station. The background and city data are then fed into a multilayer perceptron neural network algorithm to determine whether sound, vehicle counts, or day and time are the best predictors for these pollutants. The authors analyze the results using a variety of statistical tests. They ultimately determine that different variables (sound, vehicle counts, or day/time) are good predictors for different trace gases but that one variable cannot be recommended for prediction of all trace gases. They also find that these variables in conjunction with their model generally do not predict PM or PN with much to any accuracy. This work is through and the manuscript is very well-written. I recommend the below changes prior to publication.
Major comments:
Section 2.3: Can the authors provide a figure or figures that describe the neural networking process visually for readers unfamiliar with neural networking? A clarification of SOMgs would be especially helpful.
Table 4: Can the authors elaborate on what the results mean? For example, what does the relative number of hidden neurons for each pollutant mean? Is a hyperbolic tangent versus logistic sigmoid transfer function meaningful?
Figure 5: Can the authors describe why traffic has no predictive capability at all for PM1 and PN10?
Figure 6: Can the authors come up with another way to plot this figure? I am automatically looking for the maximum bar for each of the pollutant categories, when the best choice is actually the minimum bar. Can something be done to better emphasize the minimum?
In regards to rain: can the authors use the traffic cams to categorize hours in to e.g. “heavy rain”, “intermittent or light rain” or “no precipitation”, or use RH values, e.g. RH >= 90% to filter for precipitation events?
Can the authors provide some insight in to why certain variables are better predictors for certain pollutants? For example, I would assume that time is the best predictor for O3 because O3 is a photochemical product and time will have the greatest correlation with JNO2. Can the authors offer discussion for other pollutants?
Minor comments:
Line 56: The authors should define PN here where it first appears in the manuscript.
Line 178-179: This description of the time is confusing. It sounds like the time is the beginning of the current hour rather than the next hour.
Author Response
Thank you very much for the evaluation. Please see the attachment.

Reviewer 3 Report
I find the results of this manuscript interesting. The method of evaluation and the ANN development are sound.
For gaseous pollutants, the ANN gives good predictive results while for particles (PM1, PM2.5, PM10 and PN) it does not. This is in line with modelling results that are currently achieved.
With regard to the sound input data for gaseous prediction (NO2, O3), it is not as good as compared with others (traffic, time) while overall it give reasonable performance. Any comment on this ?
Finally in the discussion section, the authors compare their results with others. The results of this work is from a street canyon. It may be different if the ANN was used for a suburban street location.
Overall I rate the work as excellent.
Author Response

(The authors gave the same response as above.)
